# Bioinformatic Analysis and Machine Learning Methods in Neonatal Sepsis: Identification of Biomarkers and Immune Infiltration

**DOI:** 10.3390/biomedicines11071853

**Published:** 2023-06-28

**Authors:** Zhou Jiang, Yujia Luo, Li Wei, Rui Gu, Xuandong Zhang, Yuanyuan Zhou, Songying Zhang

**Affiliations:** 1Department of NICU, Sir Run Run Shaw Hospital, School of Medicine, Zhejiang University, No. 368 Xiasha Road, Qiantang District, Hangzhou 310016, China; 5200013@zju.edu.cn (Z.J.); yoga_luo@zju.edu.cn (Y.L.); 5192001@zju.edu.cn (L.W.); gurui7854@zju.edu.cn (R.G.); zxd123@zju.edu.cn (X.Z.); 2Department of Reproductive Endocrinology, Women’s Hospital, School of Medicine, Zhejiang University, Hangzhou 310006, China; zhouyuanyuan0624@zju.edu.cn; 3Department of Obstetrics and Gynecology, Sir Run Run Shaw Hospital, School of Medicine, Zhejiang University, No. 3 Qingchun East Road, Shangcheng District, Hangzhou 310016, China

**Keywords:** neonatal sepsis, biomarkers, immune infiltration, machine learning, GEO database

## Abstract

The disease neonatal sepsis (NS) poses a serious threat to life, and its pathogenesis remains unclear. Using the Gene Expression Omnibus (GEO) database, differentially expressed genes (DEGs) were identified and functional enrichment analyses were conducted. Three machine learning algorithms containing the least absolute shrinkage and selection operator (LASSO), support vector machine recursive feature elimination (SVM-RFE), and random forest (RF) were applied to identify the optimal feature genes (OFGs). This study conducted CIBERSORT to present the abundance of immune infiltrates between septic and control neonates and assessed the relationship between OFGs and immune cells. In total, 44 DEGs were discovered between the septic and control newborns. Throughout the enrichment analysis, DEGs were primarily related to inflammatory signaling pathways and immune responses. The OFGs derived from machine learning algorithms were intersected to yield four biomarkers, namely Hexokinase 3 (*HK3*), Cystatin 7 (*CST7*), Resistin (*RETN*), and Glycogenin 1 (*GYG1*). The potential biomarkers were validated in other datasets and LPS-stimulated HEUVCs. Septic infants showed a higher proportion of neutrophils (*p* < 0.001), M0 macrophages (*p* < 0.001), and regulatory T cells (*p* = 0.004). *HK3*, *CST7*, *RETN*, and *GYG1* showed significant correlations with immune cells. Overall, the biomarkers offered promising insights into the molecular mechanisms of immune regulation for the prediction and treatment of NS.

## 1. Introduction

Neonatal sepsis (NS) is an illness in which pathogenic or conditionally pathogenic bacteria invade the blood system during the neonatal period and produce toxins that subsequently cause systemic infection in neonates [1]. Due to its insidious manifestation in early infection, it is difficult to recognize and easy to delay treatment, which may cause septic meningitis, infectious toxic shock, or disseminated intravascular coagulation. This can lead to tissue damage and organ failure which can seriously threaten the life and health of newborns [2]. Globally, the prevalence of NS is 2%, and the overall mortality rate is about 15% [3], which ranks it as the third highest cause of neonatal death in developing nations after preterm birth and neonatal asphyxia [4]. Therefore, the early recognition of NS and timely and effective anti-infective treatment can significantly improve neonatal prognosis [1].

Neonates are immunocompromised, meaning they have an increased susceptibility to sepsis. The clinical manifestations of the disease are not obvious at the time of illness and do not allow for the early diagnosis of sepsis. Therefore, there is a need to rely on constantly evolving laboratory diagnostic techniques [5]. Currently, several blood biomarkers, comprising C-Reactive Protein (CRP), Procalcitonin (PCT), and Interleukin-6 (IL-6), have been utilized to aid in the diagnosis and prediction of NS [6,7], but there is still room for improvement when it comes to their sensitivity and specificity [8,9]. Although progress has been made in diagnosing sepsis, there is no effective method for diagnosing NS accurately and rapidly [10]. For this reason, it is imperative to explore the molecular changes involved in the pathogenesis of NS, and to discover valid new biomarkers that can help to diagnose and treat NS in its early stages.

Currently, screening for genetic alterations associated with sepsis can be accomplished by integrating bioinformatics analysis with microarray data, which can then be further applied for the early diagnosis of NS and the assessment of prognosis [11,12,13,14]. Machine learning (ML), a branch of artificial intelligence, is an advanced biometric analysis technique widely used in the medical field to diagnose diseases, predict risks, and find biomarkers. For example, Abbas et al. screened 10 key genes for an accurate diagnosis of childhood sepsis by reordering differentially expressed genes (DEGs) based on a minimum redundancy maximum correlation criteria approach using a repeated cross-validation feature selection procedure [13]. Yan et al. filtered the critical genes in NS by the least absolute shrinkage and selection operator (LASSO) algorithm and explored their correlation with immune cell infiltration [14]. Overall, it is known that the activation of innate immune cells and immune paralysis of lymphocytes are essential parts of the systemic inflammatory response in sepsis. Thus, it is becoming increasingly apparent that immune cell infiltration is essential in sepsis [15]. CIBERPORT is a popular computational method for identifying immune cells. For example, Li et al. found a significantly lower proportion of 10 immune cell types (like T cells) and a remarkably higher ratio of neutrophils and monocytes in children with sepsis compared to controls [16]. Similar results were also reported by Huang et al. in a study analyzing a population-wide, multicenter septic gene expression profile dataset [17]. However, immune function differs significantly between neonates and adults, and we only know very little about the immune cell infiltration in NS.

Therefore, in the present study, we obtained three NS datasets from the Gene Expression Omnibus (GEO) database. We used three ML methods to identify the optimal feature genes (OFGs) of NS in the training set, applied CIBERPORT to analyze immune cell infiltration, and examined biomarkers using the validation set in order to obtain novel insights into the early diagnosis and prevention of NS.

## 2. Materials and Methods

### 2.1. Collecting and Preprocessing Data

A workflow of our study is presented in Figure 1. The GSE69686, GSE25504, and GSE26440 datasets were downloaded from the GEO database [18]. GSE69686 contained 64 septic blood samples and 85 control samples. The data for the 170 neonatal samples in GSE25504 was from four platforms. The dataset based on GPL6947 had 63 samples, which consisted of 28 sepsis patients and 35 controls. And The GPL13667-based dataset had 9 sepsis, 6 healthy, 3 viral infection and 2 necrotizing enterocolitis samples, and we excluded the latter 5 samples. Then, different blocks of datasets were combined, and our batch-normalization procedure was carried out using two R packages “limma” and “sva” (v4.2.2) [19]. DEGs were identified using these datasets. Before the DEG screening analysis, a 2:1 ratio was applied using R software to randomize 101 infected and 126 control neonates to training and testing cohorts. Then, subsequent analyses were performed mainly in the training cohort. In addition, the dataset GSE26440 covered 98 children with septic shock and 32 controls. Among them, 16 infants with septic shock under 3 months of age and 10 controls were selected for validation.

### 2.2. Identification of Differentially Expressed Genes

Gene expression differences were evaluated between septic and control infants using R package “limma” with a critical value of |log_2_FC| > 1.0, as well as adjusted *p*-values < 0.05. The R packages “ggplot2” and “pheatmap” were used to generate visualizations of DEGs, including heatmaps and volcano graphs.

### 2.3. Evaluation of Functional Enrichment

Analyses and plots of Gene Ontology (GO) and the Kyoto Encyclopedia of Genes and Genomes (KEGG) pathway enrichment were analyzed with the “clusterProfiler” package in R to explore some possible biological features of DEGs. Gene set enrichment analysis (GSEA) was used to investigate the related function enrichments of all genes using the “clusterProfiler” package in R.

### 2.4. Filtering of Optimal Feature Genes

Three typical ML computations were performed to identify the OFGs. In order to select the OFGs from the DEGs, the LASSO binary logistic regression model was built using the “glmnet” package in R. The optimal penalty parameter was defined based on a 10-fold cross-validation minimum and used for each signature [20]. The OFGs were determined by applying a support vector machine recursive feature elimination (SVM-RFE) algorithm based on a nonlinear SVM using R package “e1071”, “kernlab”, and “caret” [21]. To screen the OFGs in the training dataset, the R package “randomforest” was applied with 500 trees generated for each datapoint, and a meanDecreaseGini score > 2 indicated an OFG [22]. Moreover, a Venn diagram was used to visualize the diagnostic biomarkers common to the three ML algorithms.

### 2.5. Validation of the OFGs’ Diagnostic Quality

Neonatal samples from the validation set were obtained in order to verify the screened OFGs. Box plots were created using the “ggpubr” and “ggplot2” packages to display the expression of the key OFGs in the blood of infected and control neonates. The “pROC” package was utilized to calculate receiver operating characteristic (ROC) curves, and the predictive values of OFGs were measured using the area under the curve (AUC) [23]. Therefore, OFGs were recognized as potential biomarkers with promising predictive and diagnostic power in the training and validation sets once the AUC exceeded 0.85.

### 2.6. Assessment of Immune Cell Infiltration

The differential abundances of 22 immune infiltrating cells were computed using the CIBERSORT algorithm (https://cibersort.stanford.edu/ (accessed on 10 January 2023)). Heatmaps and violin plots indicating immune cell correlation were prepared using the R packages “corrplot” and “ggplot2” [24]. The associations between OFGs and immune infiltrating cells were assessed using Spearman’s correlation coefficients, calculated using the R package “ggpubr”.

### 2.7. Culture and Stimulation of Cells

Human umbilical vein endothelial cells (HUVECs) (ATCC, Manassas, VA, USA) were used to replicate the inflammatory sepsis condition in which endothelial cells are implicated. The RPMI 1640 (Gibco, Gaithersburg, MD, USA) medium was used to cultivate HUVECs. It contains 10% fetal bovine serum and 1% penicillin–streptomycin. For passages 5–7, the cells were cultured at 37 °C, 5% CO_2_, and saturating humidity. As Mussap et al. reported, the sepsis model was established by stimulating HUVECs with LPS (1 ug/mL, Sigma, St. Louis, MO, USA) for 6 h before collecting the cells [25,26].

### 2.8. Quantitative Real-Time PCR (qRT-PCR)

The total RNA was extracted from HUVECs using an RNA-Quick Purification kit (Qiagen 74034, Hilden, Germany), and this RNA was then reverse-transcribed into cDNA using an RT-PCR Kit (A3500, Promega, Madison, WI, USA). SYBR Green Master Mix (DBI-2044, Freiberg, Germany) was mixed with cDNA for the real-time PCR. Table 1 includes a list of all primers used for the quantitative PCR. GAPDH was used as an internal control. The 2^−ΔΔCt^ method was used to calculate the expression levels of the OFGs.

### 2.9. Statistical Analysis

R version 4.2.2 was utilized for all statistical analyses and graphics. Statistical significance was defined as a *p*-value below 0.05.

## 3. Results

### 3.1. Identification of DEGs

Figure 1 demonstrates the organization of the study’s workflow. The training dataset, consisting of blood samples from 68 septic neonates and 85 control neonates, was used to analyze the DEGs. The sepsis group expressed 44 DEGs containing 41 upregulated genes and 3 downregulated genes. Following the identification of DEGs, heatmaps and volcano plots were created to present the findings (Figure 2).

### 3.2. Evaluation of Functional Enrichment

Using the R package “clusterProfiler”, DEGs enriched in the GO and KEGG pathways were examined. The results were primarily concentrated on immune and inflammatory responses. A threshold of adj. *p* value < 0.05 corrected by FDR was set, and the top five biological processes (BPs), cellular components (CCs), and molecular functions (MFs) were displayed, with the defense response to a bacterium, the secretory granule lumen, and immune receptor activity being the most enriched functions in sepsis patients (Figure 3A). The KEGG analysis uncovered a lot of immune-related enriched pathways, including the TNF signaling pathway, IL-17 signaling pathway, and cytokine–cytokine receptor interaction. The adj. *p* value < 0.2 was corrected using FDR (Figure 3B). Furthermore, a GSEA analysis revealed tight correlations of all genes with the toll-like receptor signaling pathway and complement and coagulation cascades in septic infants (Figure 3D).

### 3.3. Filtering and Validation of the OFGs

We identified eight key genes among sepsis-related DEGs using the LASSO algorithm (Figure 4A), and SVM-REF was used to select ten useful genes (Figure 4B). Additionally, the RF algorithm filtered out nine valuable genes (Figure 4C). The four upregulated OFGs, called Hexokinase 3 (*HK3*), Cystatin 7 (*CST7*), Resistin (*RETN*), and Glycogenin 1 (*GYG1*), were obtained by intersecting the genes with three advanced machine learning techniques. After further validation by the testing group, the expression of these four OFGs was significantly increased in neonates with sepsis (Figure 5). After that, we constructed ROC curves of the OFGs in the validation dataset to check their diagnostic efficiency. The results showed that in the testing group, all OFGs showed exciting diagnostic values with AUCs above 0.88 (Appendix A). In addition, an external dataset on 3-month-old infants with septic shock was selected for validation. The results showed that the AUC values for the genes were above 0.92, except for *CST7* (Figure 6). Next, we measured the expression levels of the OFGs, two down-regulated genes (*CLC* and *LRRN3*) and classic sepsis genes (*MMP8* and *MPO*) in LPS-treated HUVECs in vitro using qRT-PCR. The results of RT-PCR were consistent with the trend of bioinformatic analysis (*p* < 0.05; Figure 7).

### 3.4. Assessment of Immune Cell Infiltration

As part of the analysis, the CIBERSORT algorithm was utilized to determine the relevant proportion of each immune cell between septic and control neonates in the training dataset. A bar chart was constructed to represent each sample based on the proportions of distinct cell subpopulations (Figure 8A). The violin graph revealed that the infiltration of neutrophils, M0 macrophages, activated dendritic cells, regulatory T cells, gamma delta T cells, and plasma cells was significant in sepsis patients. Conversely, the infiltration of CD4 naive T cells, CD4 memory resting T cells, activated NK cells, and CD8 T cells showed high significance in the control group (Figure 8B). Meanwhile, the relationships between immune cells indicated that resting dendritic cells presented the most positive correlation with M2 macrophages, despite regulatory T cells having the most negative association with CD4 memory resting T cells (Figure 8B).

### 3.5. Association of Biomarkers with Infiltrating Cells

Based on a correlation analysis, we assessed the relationship between infiltrating immune cells and biomarkers. We revealed four biomarkers that were positively associated with neutrophils (*HK3*: R = 0.67; *CST7*: R = 0.65; *RETN*: R = 0.62; *GYG1*: R = 0.72, *p* < 0.001), and M0 macrophages (*HK3*: R = 0.56; *CST7*: R = 0.55; *RETN*: R = 0.51; *GYG1*: R = 0.57, *p* < 0.001), but negatively correlated with CD4 memory resting T cells (*HK3*: R = −0.61; *CST7*: R = −0.54; *RETN*: R = −0.54; *GYG1*: R = −0.58, *p* < 0.001), and CD8 T cells (*HK3*: R = −0.51; *CST7*: R = −0.56; *RETN*: R = −0.64; *GYG1*: R = −0.63, *p* < 0.001) (Figure 9). In addition, four biomarkers demonstrated positive correlations with the activation of dendritic cells, mast cells, and CD4 memory T cells, and the mobilization of regulatory T cells, naive B cells, and gamma delta T cells; however, they demonstrated negative correlations with the activation of NK cells, eosinophils, and plasma cells (Figure 9).

## 4. Discussion

Neonatal sepsis is characterized by insidious onset, high morbidity and mortality, and is complicated by distant neurological sequelae. Thus, early identification and precise treatment are key to managing this condition. In recent years, machine learning strategies have become powerful tools for studying potential relationships between high-dimensional data, and setting optimal parameters for the selection of biomarkers in biologically significant DEGs [27]. Several studies have explored sepsis biomarkers in infants and children using machine learning algorithms [13,14]. For example, Abbas et al. explored the biomarkers of childhood sepsis using three algorithms, eXtreme Gradient Boosting, RF algorithm, and logistic regression with L2 regularization. Yan et al. screened the key sepsis genes in 3-month-old infants using the LASSO algorithm. Our study applied the LASSO, SVM-RFE, and RF algorithms to screen for the prospective biomarkers of NS, and further validated their effectiveness in the test and septic shock datasets. In this study, *HK3*, *CST7*, *RETN*, and *GYG1* were screened as biomarkers of NS and had AUCs greater than 0.88 in both the test and septic shock datasets, indicating their potential predictive ability in NS and septic shock. In addition, *HK3*, *CST7*, *RETN*, *GYG1*, *MMP-8*, *MPO*, *CLC*, and *LRRN3* were also validated in LPS-stimulated HEUVCs.

Based on machine learning algorithms, our study identified *HK3*, *CST7*, *RETN*, and *GYG1* as diagnostic markers for NS. *RETN* belongs to the resistin-like molecule family and is known as a pro-inflammatory factor that inhibits reactive oxygen species production, bacterial clearance, and neutrophil migration [28]. Khattab et al. found that *RETN* levels were significantly elevated in neonates suffering from sepsis, infectious shock, or undergoing mechanical ventilation, and were positively correlated with CRP, procalcitonin, and IL-6 levels [29]. *RETN* can be used as an indicator for the diagnosis of NS, which is consistent with our results. *CST7*, specifically expressed in immune cells, is a cysteine protease inhibitor located in the cell endosomal/lysosomal vesicles; it can regulate cysteine protease activity, thus indirectly affecting the immune response process. Sawyer et al. have discovered that a high level of *CST7* was specifically expressed in circulating neutrophils from patients with sepsis [30]. Moreover, the TLR4 signaling pathway of monocyte-derived dendritic cells was activated by LPS with a significant upregulation of *CST7* expression [31]. The results indicated the involvement of *CST7* in the inflammatory process of sepsis, consistent with the findings of our study. Unfortunately, compared with classical inflammatory markers, the diagnostic value of *RETN* and *CST7* in NS still needs to be explored further and used more widely before conclusions can be made [32,33,34]. This suggests that *HK3* is involved in inflammatory and immune response processes. *GYG1* belongs to the glycogenin family and is primarily in charge of initiating glycogen synthesis. The deletion of *GYG1* results in impaired glycogen synthesis, leading to glycogen storage disease and polysomal myopathy [35,36]. In studies of sepsis, neither *HK 3* nor *GYG1* have been reported. Their roles in the metabolic and immune processes of neonatal sepsis need to be investigated further.

Sepsis-induced innate immune dysregulation and adaptive immune restriction, in combination, trigger sustained pro- and anti-inflammatory pathways that ultimately lead to tissue and organ dysfunction [15]. In our study, to assess the impact of the immune system on NS more comprehensively, patients were assessed for immune infiltration using CIBERSORT. The results showed a greater ratio of neutrophils, M0 macrophages, and regulatory T cells, and a lower percentage of CD4 naive T cells, activated NK cells, CD8 T cells, and CD4 memory resting T cells in septic neonates, suggesting that these cells are associated with the development and progression of NS. Neutrophils are engaged in the early acute inflammatory response, primarily through release, migration, and phagocytosis [37]. In neonates, neutrophils remain the most important defense barrier in response to sepsis despite their insufficient number. In *CEBP* knockout *mice*, neutrophil production was impaired, which increased the risk of opportunistic infections and death [38]. The levels of regulatory T cells (Tregs) were significantly raised in septic neonates and were positively associated with mortality. Simvastatin treatment, or Rho kinase activity inhibition, reduced Treg levels in CLP *mice* and improved the prognosis of sepsis [39,40]. However, the low activity of NK cells in neonates increases their susceptibility to sepsis. After the onset of sepsis, the activity and toxicity of NK cells were further reduced, leading to disease progression and adverse outcomes [41]. In addition, PD-1 exerted an immunosuppressive effect in NS by downregulating CD8+ T cell activity during the inflammatory response and exacerbating disease progression, due to the acquired immune deficiency of neonates [42]. However, the association of CD4 T cells with NS has not been reported and thus requires further investigation. In addition, we investigated the correlation between biomarkers and infiltrating immune cells. The findings revealed significant correlations between all four biomarkers and neutrophils, M0 macrophages, CD4 naive T cells, and CD8 T cells. It was shown that upregulated *CST7* is a specific hallmark of neutrophils in septic infants [31]. The pro-inflammatory factor *RETN* can directly inhibit the bactericidal effect of neutrophils in septic infants and promote the progression of inflammation [28]. However, more information is needed on the complex interaction processes between key genes and immune cells. Based on these findings, it is imperative that we investigate the potential molecular mechanisms and functional implications of immune cell infiltration in NS.

In the present study, we applied the LASSO, SVM-RFE, and RF algorithms to identify four optimal feature genes in NS patients with reduced bias. Then, three datasets from different platforms were merged, batch corrected, and randomly grouped to improve the reliability of the statistical analysis. In addition, the diagnostic effect of these genes on septic shock was validated by our study. However, this study also had some limitations. Due to the limited number of samples used, more clinical samples are required in order to validate and evaluate the reliability of our results. Additionally, the NS-related molecular mechanisms should be investigated further by constructing animal models and cellular experiments.

## 5. Conclusions

By combining the optimal feature genes of three machine learning algorithms, we identified four potential biomarkers—*HK3*, *CST7*, *RETN*, and *GYG1*—and discovered a close correlation between these biomarkers and immune cell infiltration. This result not only provides the ability to diagnose NS early, but also introduces new insights into the immunotherapy of NS patients.

## Figures and Tables

**Figure 1 biomedicines-11-01853-f001:**
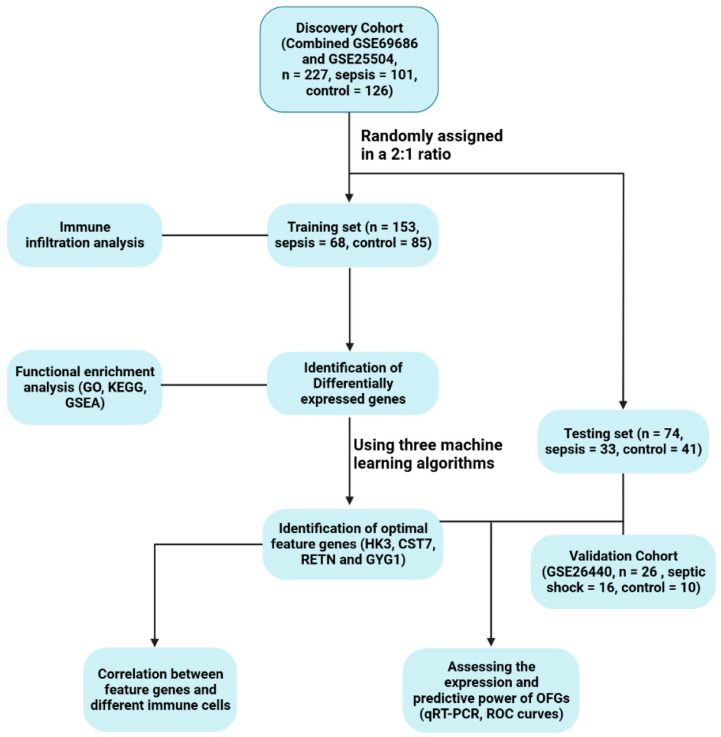
The flow diagram of the study.

**Figure 2 biomedicines-11-01853-f002:**
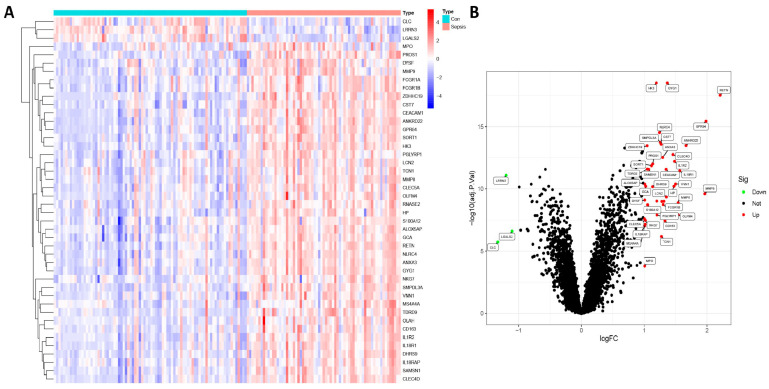
Genes differentially expressed between Neonatal Sepsis and Controls (Con). (**A**) Heatmaps and (**B**) Volcano plots.

**Figure 3 biomedicines-11-01853-f003:**
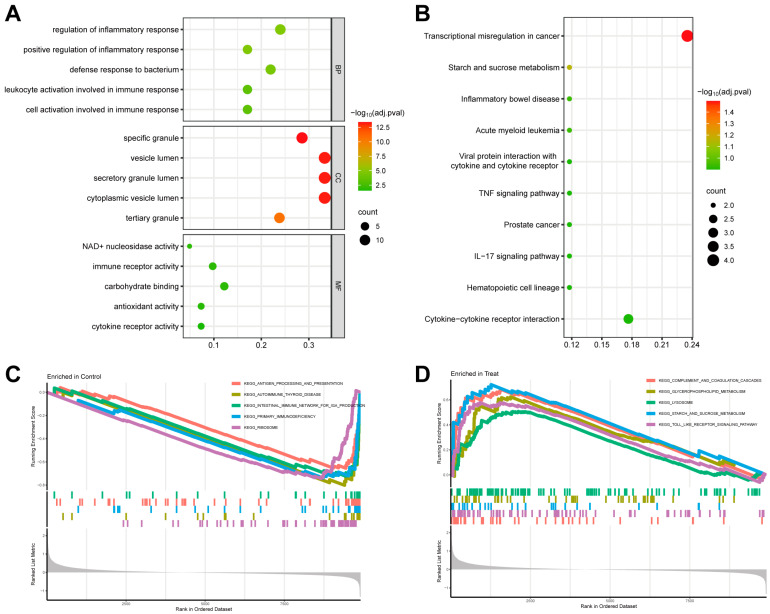
Evaluation of functional enrichment. (**A**) GO analyses were performed to forecast the prospective functions of DEGs, containing CC, BP, and MF. (**B**) KEGG pathways concerning DEGs were assessed. (**C**,**D**) GSEA exhibited 5 critical signaling pathways that were enriched in control and NS groups, respectively.

**Figure 4 biomedicines-11-01853-f004:**
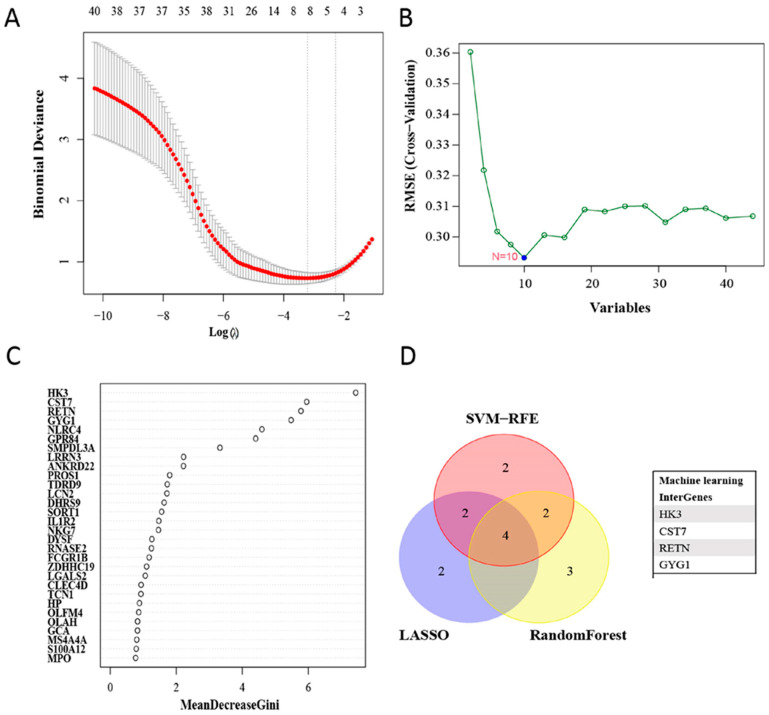
Filtering potential OFGs of Neonatal Sepsis. (**A**) Identified OFGs through LASSO algorithm. (**B**) Selected OFGs by SVM-RFE algorithm. (**C**) RF algorithm select OFGs with the MeanDecreaseGini score > 2. (**D**) Venn diagram showing four diagnostic biomarkers intersected by three algorithms.

**Figure 5 biomedicines-11-01853-f005:**
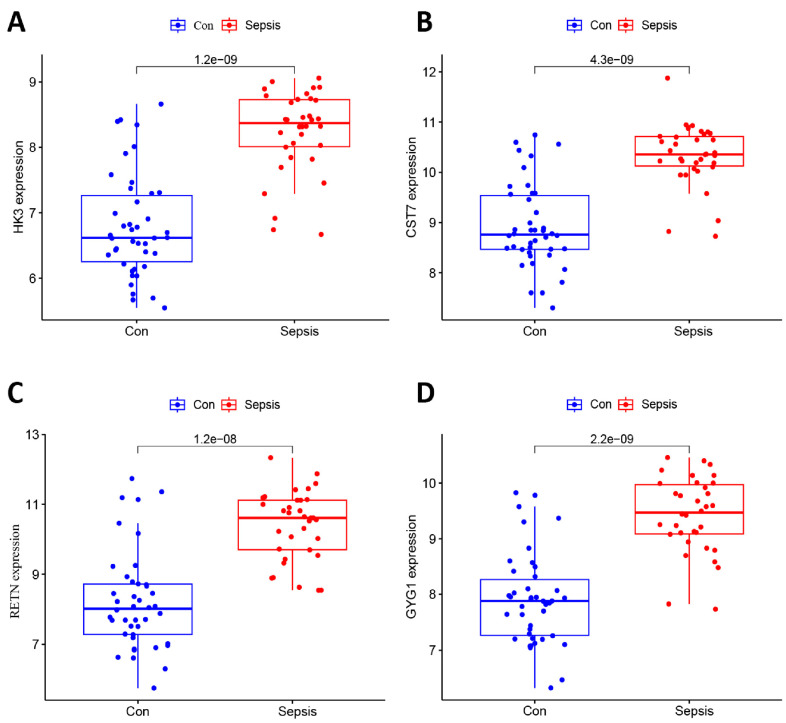
Validation of the OFGs in testing dataset. (**A**–**D**) Expression of *HK3*, *CST7*, *RETN,* and *GYG1* in testing dataset of septic and control (Con) neonates.

**Figure 6 biomedicines-11-01853-f006:**
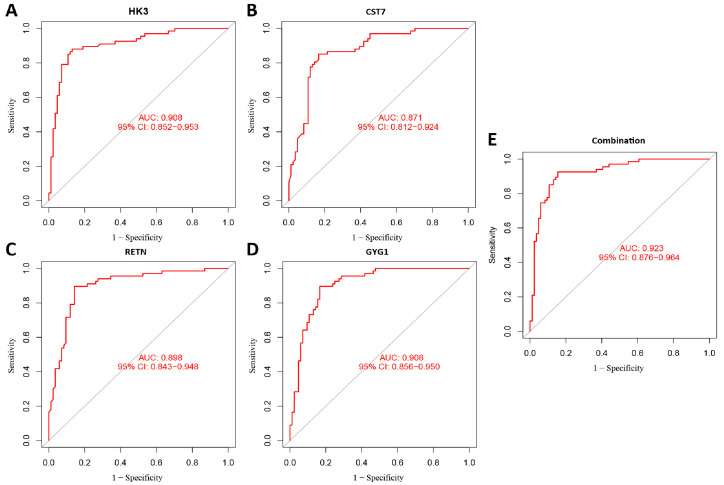
Diagnostic effectiveness of OFGs in the dataset of septic shock. (**A**–**E**) ROC curves for diagnostic effectiveness of *HK3*, *CST7*, *RETN*, *GYG1*, and combined in neonatal septic shock.

**Figure 7 biomedicines-11-01853-f007:**
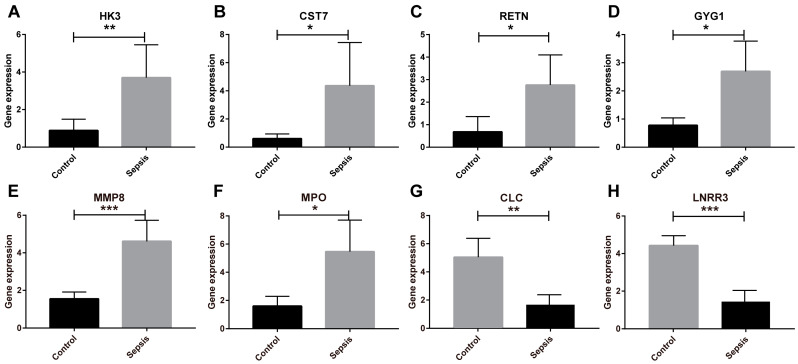
Validation in LPS-treated HUVECs. The expression levels of the OFGs were higher in LPS-treated HUVECs. (**A**) *HK3*. (**B**) *CST7*. (**C**) *RETN*. (**D**) *GYG1*. (**E**) *MMP-8*. (**F**) *MPO*. (**G**) *CLC*. (**H**) *LRRN3*. Notes: * *p* < 0.05, ** *p* < 0.01, *** *p* < 0.001.

**Figure 8 biomedicines-11-01853-f008:**
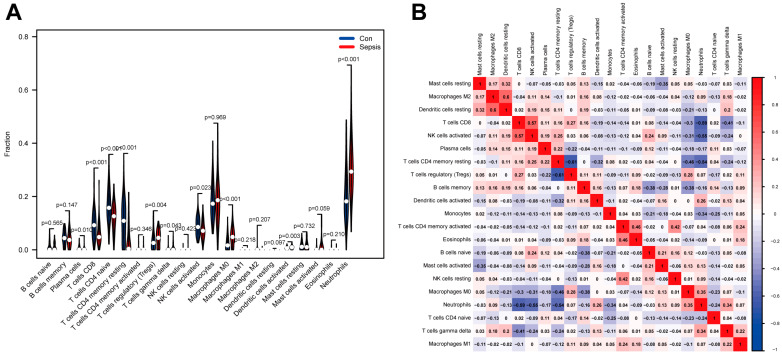
Compared and correlated in immune cell infiltration. (**A**) Violin graph shows the distinct fractions compared to each immune cell in NS and control (Con) groups. (**B**) Analysis of correlations among 22 immune cell subtypes.

**Figure 9 biomedicines-11-01853-f009:**
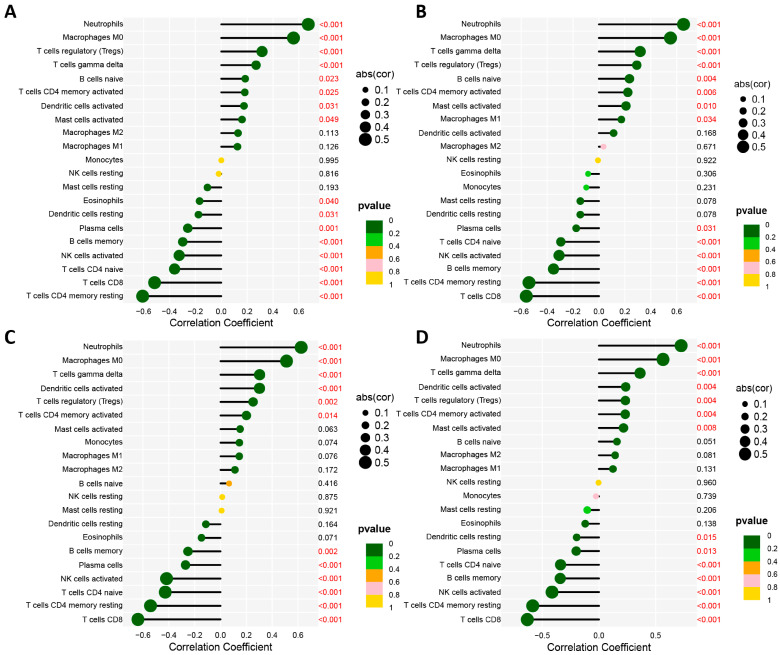
Visualization of Spearman correlation between 4 optimal feature genes and immune infiltrating cells in neonatal sepsis (NS). (**A**) *HK3*. (**B**) *CST7*. (**C**) *RETN*. (**D**) *GYG1*.

**Table 1 biomedicines-11-01853-t001:** Primers for quantitative real-time PCR.

Primers	Sequence (5′→3′)	
*HK3*	Forward	5′-GGACAGGAGCACCCTCATTTC-3′
	Reverse	5′-CCTCCGAATGGCATCTCTCAG-3′
*CST7*	Forward	5′-GTGTGAAGCCAGGATTTCCTAA-3′
	Reverse	5′-TGTCGTTCGTGCAGTTGTTGA-3′
*RETN*	Forward	5′-CTGTTGGTGTCTAGCAAGACC-3′
	Reverse	5′-CCAATGCTGCTTATTGCCCTAAA-3′
*GYG1*	Forward	5′-TGACACTAACCACAAACGATGC-3′
	Reverse	5′-TAGATGAGCAGAATCGCCACT-3′
*CLC*	Forward	5′-TCTACTGTGACAATCAAAGGGC-3′
	Reverse	5′-CACGACGACCAAAGCACAC-3′
*LRRN3*	Forward	5′-AAGCCTCTTATCAATCTTCGCAG-3′
	Reverse	5′-CCAGTCCAACCAAGGCGTTA-3′
*MMP8*	Forward	5′-GGAAGGCAGGAGAGGTTGTC-3′
	Reverse	5′-GTTGAAAGGCATGGGCAAGG-3′
*MPO*	Forward	5′-TTGACAACCTGCACGATGAC-3′
	Reverse	5′-TGTGCTCCCGAAGTAAGAG-3′
*GAPDH*	Forward	5′-GCCTCAAAATCCTCTCGTTGTG-3′
	Reverse	5′-GGAAGATGGTGATGGGATTTC-3′

## Data Availability

The datasets supporting the conclusions of this article are available in the GEO database (http://www.ncbi.nlm.nih.gov/geo (accessed on 5 January 2023)).

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
