# Peer review of "Bioinformatic Analysis and Machine Learning Methods in Neonatal Sepsis: Identification of Biomarkers and Immune Infiltration"

_biomedicines, 2023, doi:10.3390/biomedicines11071853_

Round 1

Reviewer 1 Report

Neonatal sepsis (NS) is a severe disease with unclear causes. This study used gene expression data to identify DEGs and performed functional enrichment analyses. Machine learning algorithms were used to select optimal genes. The study also analyzed immune cell infiltrates in septic infants and identified four potential biomarkers (HK3, CST7, RETN, and GYG1). These biomarkers showed correlations with immune cells. The authors share insights on the molecular mechanisms of NS.

Major comments:

1.     Data selection. The use of GSE69686, GSE25504, and GSE26440 datasets from GEO is legitimate but not correctly justified. For example, GSE26440 provides 130 samples (with different annotation referring to survival etc). This study covers 98 children with septic shock and 32 controls. Each of these GEO is supported by a detailed publication that deals with finding and biomarkers that best applied to the experiments. The authors mention that 9 sepsis and 6 control samples were used. What is the basis for this selection? This comment applies for all input data.  

2.     Missing references for the accumulated knowledge on HUVECs and sepsis (e.g., Mussap et al 2013; Pietrasanta et al 2019; Fu et al 2020).

3.     KEGG/ GO enrichment (Figure 3) is only valid by providing the real statistical finding (unselected, raw data). The authors mention <0.05 but it is not clear what form of correction done on the raw p-value (e.g., Bonferroni? FDR).

4.     Missing supplementary tables for the results of the 3 machine learning tests (i.e., the gene list as the source for the Venn diagram). The reader should be able to see the full list, the thresholds used for each ML method etc.

5.     Include RT-PCR analysis for some genes that are NOT in the selected 4-gene list. Whether these are genes that were previously indicated (e.g. MMP-8 which was upregulated in sepsis) or by your ranked list (as in Fig 5).

Minor comments:

1.     Fig. 1 can be compress to the 8 sections (if needed for each section mention the relevant Figure number. The visualization of the results is useless (exception is the scheme of the work in 1). No need to mention R code etc).

2.     Avoid confusion: “We also collected blood samples …” it seems that you have used public data and you have not collected the samples. Please clarify.

3.     How sensitive are the results to the arbitrary decision to use of 2:1 ratio for train-test cohorts.

4.     Which correlation is shown in Fig. 9C? Fig 9A is not quantitative and it is too detailed. Can be removed.

5.     Avoid over interpretation (e.g. cancer setting is too speculative-lines 277-280).

6.     Fig 6 should move to supplemental data. Add table (to supplement) with specific performance of the gene-models (F1 value, recall, sensitivity).

can benefit from English editing

Author Response

Dear reviewer,

Once again, we want to express our appreciation. Thank you so much for your excellent comments and creative suggestions on our manuscript. We have read your comments carefully and revised portion are marked in yellow in the paper. Please see the attachment with the main corrections in the paper and the responds to the reviewer’s comments.

Reviewer 2 Report

Review for the manuscript “Bioinformatic analysis and machine learning methods in 2 neonatal sepsis: identification of biomarkers and immune 3 infiltration”.

Neonatal sepsis (NS), a public health problem, represents one of the common diseases in the neonatal intensive care unit. It is associated with higher morbidity and mortality.

In the present study, the authors explored sepsis biomarkers using LASSO, SVM-RFE and RF algorithm. Using machine learning algorithms, their study identified HK3, CST7, RETN and GYG1 as diagnostic markers for NS.

They also, validated the diagnostic effect of these genes on septic shock.

The results are very interesting and important for neonatologists.

 The paper has higher level of novelty. I also remarked the quality of presentation with one table and 10 figures and also the high interest to the readers.

Author Response

Dear reviewer,

Once again, we want to express our appreciation. Thank you so much for your excellent comments and creative suggestions on our manuscript.

Response 1: Thank you very much for your review.

Reviewer 3 Report

This paper represents the first successful attempt to determine a diagnostic signature for sepsis in neonates by combining three contemporary, powerful machine learning bioinformatics algorithms, LASSO, SVM-RFE and RF.  Despite several papers relative to gene expression signature studies in adult sepsis patients, no relevant studies has been performed so far in neonate cohorts by combining all available gene expression data. The merit of the paper is closing a gap in sophisticated, combined  analysis of DEG from 101 infected and 126 control neonates subdivided into  training and testing cohorts.The undoubted advantage of the study is defining a proof for a neonatal sepsis diagnostic signature comprised of  4 upregulated genes (HK3, CST7, RETN, and GYG1 . Most importantly, data demonstrate highly informative ROC curves (0.92-0.99) for this signature in testing dataset and separate septic shock specimens. Figures and explanations are ivery informative.

Line 65: activation of congenital immune cells 

 Manuscript requires light editing (line 65: "activation of congenital immune cells" is not correct phrase; instead, "innate immunity cells" look as a preferable commonly used term ). 

Author Response

Dear reviewer,

Once again, we want to express our appreciation. Thank you so much for your excellent comments and creative suggestions on our manuscript. We have read your comments carefully and revised portion are marked in yellow in the paper. The main corrections in the paper and the responds to the reviewer’s comments are as following:

Point 1: Comments on the Quality of English Language: Manuscript requires light editing (line 65: "activation of congenital immune cells" is not correct phrase; instead, "innate immunity cells" look as a preferable commonly used term).

Response 1: Thank you very much for your suggestion. We have made modifications to the abstract, which has marked in "Introduction" on page 2, line 72. And we have made a language editing by MDPI. The certificate has been uploaded to the attachment.

Round 2

Reviewer 1 Report

1. In general, the authors better explain the selection of the data and improve Fig 1. 

2. Please remove Fig 9A that is non informative (you mention that it was removed, but I still see it).

3. In Fig 3  - add legend to x-axis. The fonts of Fig 3C and 3D are too small (also do not use the word 'treated' )

Only minor corrections is needed